# Transferring Optimality Across Data Distributions via Homotopy Methods

**Matilde Gargiani**[1], **Andrea Zanelli**[2], **Quoc Tran-Dinh**[3], **Moritz Diehl**[2,4], **Frank Hutter**[1,5]

[1]Department of Computer Science, University of Freiburg
`{gargiani, fh}@cs.uni-freiburg.de`
[2]Department of Microsystems Engineering (IMTEK), University of Freiburg
`{andrea.zanelli, moritz.diehl}@imtek.uni-freiburg.de`
[3]Department of Statistics and Operations Research, University of North Carolina
`quoctd@email.unc.edu`
[4]Department of Mathematics, University of Freiburg
[5]Bosch Center for Artificial Intelligence

## Abstract

Homotopy methods, also known as continuation methods, are a powerful mathematical tool to efficiently solve various problems in numerical analysis. In this work, we propose a novel homotopy-based numerical method that can be used to gradually transfer optimized parameters of a neural network across different data distributions. This method generalizes the widely-used heuristic of pre-training parameters on one dataset and then fine-tuning them on another dataset of interest. We conduct a theoretical analysis showing that, under some assumptions, the homotopy method combined with Stochastic Gradient Descent (SGD) is guaranteed to converge in expectation to an $r_\theta$-optimal solution for a target task when started from an expected $r_\theta$-optimal solution on a source task. Empirical evaluations on a toy regression dataset and for transferring optimized parameters from MNIST to Fashion-MNIST and CIFAR-10 show substantial improvement of the numerical performance over random initialization and pre-training.

## 1 Introduction

Homotopy methods (Allgower & Georg, 1980), also known as continuation methods, are a powerful mathematical tool to efficiently solve various problems in numerical analysis (e.g., Tran-Dinh et al. (2012), Zanelli et al. (2019)). The core idea consists in sequentially solving a series of parametric problems, starting from an easy-to-solve problem and progressively deforming it, via a homotopy function, to the target one. Homotopy methods are suitable to solve complex non-convex optimization problems where no or only little prior knowledge regarding the localization of the solutions is available. In addition, in contrast to state-of-the-art algorithms in deep learning (e.g., Bottou (2010), Duchi et al. (2011), Kingma & Ba (2015)), these methods often achieve global convergence guarantees by only exploiting local structures of the problem. Concepts, such as curriculum-learning and warm-starting, that are related to different degrees to homotopy methods, have been explored both in the deep learning (e.g., Gulcehre et al. (2016), Mobahi (2016), Gulcehre et al. (2017)) and in the reinforcement learning (e.g., Narvekar (2017)) communities.

In this work, we propose a novel homotopy-based numerical method to transfer knowledge regarding the localization of a minimizer across different task distributions in deep learning. This method gradually tracks a neural network's (close-to-)optimal parameters from one data distribution to another one via the homotopy method (Allgower & Georg, 1980) and can be interpreted as a generalization of the very common heuristic of fine-tuning a pre-trained network. After discussing related work (Section 2) and background on homotopy methods (Section 3), our contributions are as follows:

1. We provide a general theoretical analysis of the homotopy method when using SGD as an iterative solver, proving that under some local assumptions it tracks in expectation an $r_\theta$-optimal solution from the source task to the target task (Section 4).

2. We introduce homotopy functions for transferring optimality across data distributions for supervised regression and classification tasks (Section 5).

3. For a toy regression dataset and for transferring optimized parameters from MNIST to Fashion-MNIST and from MNIST to CIFAR-10, we show that our method obtains up to two orders of magnitude better numerical performance than random initialization and substantial improvement of the numerical performance over pre-training (Section 6).

## 2 RELATED WORK

Deep neural networks have led to establish a new state-of-the-art in many applications. Despite their great success and the many theoretical studies that have been published in the last years (e.g., Balduzzi et al. (2017), Li et al. (2018), Feizi et al. (2018), Kunin et al. (2019)), training these deep models remains a big challenge. Various stochastic optimization algorithms (e.g., Duchi et al. (2011), Kingma & Ba (2015), Reddi et al. (2018)) and initialization heuristics (e.g., Daniely et al. (2016), Klambauer et al. (2017), Hanin & Rolnick (2018)) have been recently suggested in order to improve and speed up the training procedure. We now briefly discuss the state-of-the-art deep learning optimization techniques and initialization strategies that are most related with the proposed homotopy-based method, drawing connections with existing and ongoing research works in the field.

**Curriculum Learning.** First introduced by Bengio et al. (2009) and then extended in different works (e.g., Graves et al. (2017), Weinshall et al. (2018), Hacohen & Weinshall (2019)), *curriculum learning* can also be listed among the optimization heuristics proposed to alleviate the complexity of solving high dimensional and non-convex problems. In particular, taking inspiration from the fact that humans and animals learn "better" when exposed to progressively more complex situations in an organized manner, curriculum learning techniques guide the training by starting with "easy-to-learn" samples and progressively introducing more "complex-to-learn" ones. This guided learning process can also be rephrased in a homotopy-like fashion (see Algorithm 1) as solving a sequence of optimization problems where the target training distribution gradually changes from considering only the "easy" examples to the full original training distribution.

**Meta-Learning and Transfer-Learning.** Due to the massive amount of computational resources required by the development of modern deep learning applications, the community has started to explore the possibility of re-using learned parameters across different tasks, leading to the development of many new *transfer-learning* (e.g., Rohrbach et al. (2013), Wang & Schneider (2014), Cui et al. (2019)) and *meta-learning* (e.g., Schmidhuber (1987), Hochreiter et al. (2001), Finn et al. (2017), Zintgraf et al. (2019)) algorithms. The simplest way to transfer knowledge across different tasks consists in using *warm-start* initialization. This heuristic is amply used in computer vision applications, where it is also known as the *fine-tuning* technique (e.g., Krizhevsky et al. (2012), Yosinski et al. (2014), Reyes et al. (2015), Käding et al. (2016)). So far, there is no rigorous explanation of why and when fine-tuning works. However, numerous empirical evaluations on different benchmarks show that warm-starting the parameters of deep models often leads to faster convergence and better generalization than using random initialization.

## 3 BACKGROUND

In this work, we will focus on solving problems of the form

$$\theta^* \in \arg\min_{\theta \in \mathbb{R}^d} \underbrace{\frac{1}{N} \sum_{j=1}^{N} \ell_j(\theta)}_{:= J(\theta)}, \tag{1}$$

where $J : \mathbb{R}^d \to \mathbb{R}$ is our target objective function and $\theta^*$ is a minimizer. Problems as described in (1) arise, for instance, in classification and regression scenarios.

In the following section we briefly review the main concepts of *homotopy* and *continuation methods*, which the proposed technique to solve problem (1) is based on.

### 3.1 Homotopic Functions and Continuation Methods for Optimization

Given two topological spaces $Z$ and $Y$, a *homotopy* is a continuous deformation between two continuous functions $g, f : Z \to Y$ that fulfills certain properties. We can formalize this concept with the following definition

**Definition 3.1.** *Let $g, f : Z \to Y$ be continuous maps on the topological spaces $Z, Y$. A homotopy from $g$ to $f$ is a continuous function $H : Z \times [0, 1] \to Y$ such that*

$$H(z, 0) = g(z), \qquad H(z, 1) = f(z), \qquad \forall z \in Z. \tag{2}$$

*If such function $H$ exists, $g$ is said to be homotopic of $f$, and this relation is denoted by $g \simeq f$.*

It is straightforward to show that, $A \subseteq \mathbb{R}^n$ being a convex set, any two continuous maps $g, f : Z \to A$ are homotopic (see (Suciu, 2016) for a derivation). From this fact it follows that any two continuous and real functions are homotopic. See Figures 4a– 4b in the appendix for a graphical representation of two different homotopy maps between the probability density functions of two Gaussian distributions, where $\lambda \in [0, 1]$ denotes the homotopy parameter. See also Section A in the appendix for details on some of the main properties of homotopic functions.

Continuation methods (also known as homotopy methods) are a widely used mathematical tool to solve complex non-convex optimization problems where no or only very limited prior knowledge regarding the localization of optimal solutions is available (see (Allgower & Georg, 1980) for a full characterization of continuation methods). The core idea of a homotopy approach consists in defining a homotopy function $H(\theta, \lambda)$ with $\lambda \in [0, 1]$ such that $H(\theta, 0) = J_0(\theta)$ is a trivial to optimize smooth map (or a smooth map of which a surrogate $\theta_0$ of an optimal solution is available) and $H(\theta, 1) = J(\theta)$ is our target objective function. Instead of directly addressing problem (1), we approximately and sequentially solve $\gamma > 0$ parametric optimization problems of the form

$$\theta_i^* \in \arg\min_{\theta \in \mathbb{R}^d} \underbrace{\frac{1}{N} \sum_{j=1}^{N} \ell_j(\theta, \lambda_i)}_{:= H(\theta, \lambda_i)}, \tag{3}$$

for increasing values of the parameter $\lambda_i$ for $i = 1, \dots, \gamma$ and warm-starting each problem with the previously derived approximate solution. Conceptually, Algorithm 1 describes the basic steps of a general homotopy algorithm. Under appropriate assumptions, if the increment $\Delta\lambda$ is sufficiently small, then the iterative procedure in Algorithm 1 will converge to a neighborhood of an optimal solution of the target objective $J$ that depends in some sense on the number of iterations $k > 0$ performed (Allgower & Georg, 1980). Many different variations of Algorithm 1 exist. In particular,

---

**Algorithm 1** A Conceptual Homotopy Algorithm

---

1: $\theta_0 \approx \theta_0^* \in \arg\min_\theta H(\theta, 0)$
2: $\gamma > 0, \gamma \in \mathbb{Z}$
3: $\lambda_0 = 0, \ \Delta\lambda = 1/\gamma$
4: $k > 0, k \in \mathbb{Z}$
5: **for** $i = 1, \dots, \gamma$ **do**
6: $\quad \lambda_i \leftarrow \lambda_{i-1} + \Delta\lambda$
7: $\quad$ **procedure** $\theta_i \leftarrow$ IterativeSolver($\theta_{i-1}, k, H(\theta, \lambda_i)$)
8: **return** $\theta_\gamma$

---

different update schemes for the homotopy parameter can be adopted (e.g., geometric or sublinear rate of increase), various iterative solvers can be used under distinct and specific assumptions, and, finally, also diverse levels of approximation for the solutions $\theta_i^*$ can be considered, i.e. different $k$ values.

Before going into the details of two concrete formulations of the conceptual homotopy method outlined in Algorithm 1 (see Section 5) when applied to transfer optimality knowledge in regression and classification scenarios, we provide a general theoretical analysis in a simplified setting.

## 4 THEORETICAL ANALYSIS

In this section, we provide a local theoretical analysis of homotopy methods when Stochastic Gradient Descent (SGD) (Bottou, 2010) is used as iterative solver in Algorithm 1. The locality of the analysis consists in the definition of hyperspheres of radius $B \geq 0$ around the optimal solutions of each homotopy problem $H(\theta, \lambda_i)$ where it is possible to exploit certain structures of the problem. In particular, we approximately and sequentially solve $\gamma > 0$ unconstrained optimization problems of the form

$$\theta_i^* \in \arg\min_{\theta \in \mathbb{R}^d} H(\theta, \lambda_i), \quad \forall i = 1, \ldots, \gamma, \tag{4}$$

where $H(\theta, \lambda_i)$ fulfills the assumptions described in Section 4.1 and $\lambda_i \in [0, 1]$. Let $\theta_i$ be an approximate solution of the problem associated with parameter $\lambda_i$ derived by applying $k > 0$ iterations of SGD (in the limit, $k = 1$) and also the starting point for the problem associated with parameter $\lambda_{i+1}, \forall i = 1, \ldots, \gamma - 1$. In addition, let $\theta_0$ denote an approximate solution for the source task, i.e. $\lambda_0 = 0$, that is used as initial point for the problem associated with $\lambda_1$. In this section we characterize the maximum allowed variation of the homotopy parameter in order for the method to able to track in expectation an $r_\theta$-optimal solution from source to target task.

### 4.1 ASSUMPTIONS

We now expose the fundamental assumptions for our general local theoretical analysis on which all the derivations in Sections 4.2 and 4.3 rely. In addition, throughout the analysis the $\ell$-functions in (3) are implicitly assumed to be differentiable in $\theta$. We start by giving the definition of the regions around the optimal solutions of the homotopy problems where the analysis is conducted.

**Definition 4.1.** *Given $\theta_i^*$ and $B \geq 0$, let $\mathcal{B}_{B,\theta_i^*}$ be the following set of vectors*

$$\mathcal{B}_{B,\theta_i^*} := \{\theta \text{ s.t. } \|\theta - \theta_i^*\| \leq B\}, \ \forall i = 0, \ldots, \gamma.$$

**Assumption 4.2** (local $L$-smoothness). *Assume that there exists a constant $L > 0$ such that*

$$\|\nabla_\theta H(\tilde{\theta}, \lambda_i) - \nabla_\theta H(\hat{\theta}, \lambda_i)\| \leq L\|\tilde{\theta} - \hat{\theta}\|, \quad \forall \tilde{\theta}, \hat{\theta} \in \mathcal{B}_{B,\theta_i^*}, \ \forall i = 0, \ldots, \gamma. \tag{5}$$

**Corollary 4.2.1.** *If $H$ is locally L-smooth in $\theta$, then the following inequality holds*

$$H(\theta_i^*, \lambda_i) - H(\hat{\theta}, \lambda_i) \leq -\frac{1}{2L}\|\nabla_\theta H(\hat{\theta}, \lambda_i)\|^2, \quad \forall \hat{\theta} \in \mathcal{B}_{B,\theta_i^*}, \ \forall i = 0, \ldots, \gamma. \tag{6}$$

*Proof.* See Lemma 1.1 in (Gower, 2018) for a proof. $\qquad\square$

**Assumption 4.3** (local $\mu$-strong convexity). *Assume that there exists $\mu > 0$ such that*

$$H(\tilde{\theta}, \lambda_i) \geq H(\hat{\theta}, \lambda_i) + \nabla_\theta H(\hat{\theta}, \lambda_i)^T(\tilde{\theta} - \hat{\theta}) + \frac{\mu}{2}\|\tilde{\theta} - \hat{\theta}\|^2, \quad \forall \tilde{\theta}, \hat{\theta} \in \mathcal{B}_{B,\theta_i^*}, \ \forall i = 0, \ldots, \gamma. \tag{7}$$

**Assumption 4.4** (bounded $\ell$-derivative). *Assume that there exists $\nu > 0$ such that*

$$\|\nabla_\theta \ell_j(\hat{\theta}, \lambda_i)\| \leq \nu, \quad \forall \hat{\theta} \in \mathcal{B}_{B,\theta_i^*}, \ \forall i = 0, \ldots, \gamma, \ \forall j = 1, \ldots, N. \tag{8}$$

**Assumption 4.5** (local bounded "variance"). *Let $g(\hat{\theta}, \lambda_i)$ denote an unbiased estimate of the gradient $\nabla_\theta H(\hat{\theta}, \lambda_i)$. Assume that there exists a constant $C \geq 0$ such that the following bound on the expected squared norm of the estimate of the gradient holds*

$$\mathbb{E}\left[\|g(\hat{\theta}, \lambda_i)\|^2\right] \leq C^2, \quad \forall \hat{\theta} \in \mathcal{B}_{B,\theta_i^*}, \ \forall i = 0, \ldots, \gamma. \tag{9}$$

**Remark 4.6.** *Assumption 4.5 is standard for proving error bounds on SGD iterates (see (Schmidt, 2014)). In addition, notice that, since*

$$\mathbb{E}\left[\|g(\hat{\theta}, \lambda_i)\|^2\right] = Var\left(\|g(\hat{\theta}, \lambda_i)\|\right) + \mathbb{E}\left[\|g(\hat{\theta}, \lambda_i)\|\right]^2,$$

*the $C$ constant is proportional to the variance and the squared expected value of the norm of the gradient estimate. Therefore, it decreases when the iterates approach a minimizer and by reducing the noise in the estimate of the gradient. In the limit (i.e. exact gradient and convergence to a minimizer), $C = 0$.*

Recall that $\theta^*(\lambda_i) \equiv \theta_i^*$.

**Assumption 4.7** (strong regularity)**.** *Assume that there exists $\delta > 0$ such that the following inequality holds*

$$\|\theta^*(\lambda_{i+1}) - \theta^*(\lambda_i)\| \leq \delta |\lambda_{i+1} - \lambda_i|\,, \quad \forall i = 0, \ldots, \gamma - 1.$$

**Remark 4.8.** *Assumption 4.7 follows directly from the application of the Implicit Function Theorem by introducing some milder assumptions on the problem structure (see Lemma 2.1.8 in (Allgower & Georg, 1980)).*

## 4.2 FUNDAMENTAL THEORETICAL PRELIMINARIES

Before proceeding with the main theoretical contributions, we extend the existing results in the literature on global error bounds for the iterates of Stochastic Gradient Descent such that they can be applied when the underlying assumptions are only required to hold locally. The derived local error bounds for SGD iterates are used in Proposition 4.11 and Theorem 4.12.

**Proposition 4.9.** *Let $\theta_i \in \mathcal{B}_{B,\theta_i^*}$ be the starting point for the problem described in (3), and let $\theta_i := \theta_{i,0}$ and $\theta_{i+1} := \theta_{i,k}$ denote the iterate after $k > 0$ SGD steps, where an SGD step is defined as*

$$\theta_{i,k} = \theta_{i,k-1} - \alpha g(\theta_{i,k-1}, \lambda_i)\,.$$

*Under Assumptions 4.2– 4.5 and by setting the batch size $0 < M \leq N$ to a value such that $\frac{(N-M)}{N} \leq \frac{(1-\kappa_d)}{2\alpha\nu} B$ with $\kappa_d = \sqrt{(1-\alpha\mu)}$ and the learning rate $\alpha$ to a constant value such that $0 < \alpha \leq \min\left(\frac{1}{2\mu}, \frac{1}{L}\right)$, the following error bound on the iterates holds*

$$\mathbb{E}\left[\|\theta_{i+1} - \theta_{i+1}^*\|^2\right] \leq (1 - 2\alpha\mu)^k \cdot \mathbb{E}\left[\|\theta_i - \theta_{i+1}^*\|^2\right] + \frac{\alpha C^2}{2\mu}\,. \tag{10}$$

*Proof.* See Section D in the appendix. $\qquad\square$

**Remark 4.10.** *The expectation in (10) is taken w.r.t. all the random variables, i.e. estimates of the gradients and initial point $\theta_0$, involved in the optimization procedure up to the current $i+1$ iteration of the algorithm.*

## 4.3 MAIN THEORETICAL CONTRIBUTIONS

Under the considered assumptions and by exploiting the previously derived results on local error bounds for SGD iterates, we show that, if the approximate solution $\theta_i$ for the problem with parameter $\lambda_i$ is "sufficiently close" to a minimizer $\theta_i^*$ in expectation, i.e. $\mathbb{E}\left[\|\theta_i - \theta_i^*\|^2\right] \leq r_\theta^2$, then, for a "sufficiently small" change in the homotopy parameter, the same vicinity to a minimizer $\theta_{i+1}^*$ is preserved in expectation for the approximate solution $\theta_{i+1}$ of the problem with parameter $\lambda_{i+1}$, i.e. $\mathbb{E}\left[\|\theta_{i+1} - \theta_{i+1}^*\|^2\right] \leq r_\theta^2$. In particular, with Theorem 4.12 we characterize the maximum allowed variation of the homotopy parameter based on the properties of the parametric problems and the convergence characteristics of the adopted iterative solver, i.e. rate of convergence and number of iterations.

First, in order to apply the results derived in Theorem 4.12, given a realization of $\theta_i \in \mathcal{B}_{B,\theta_i^*}$, we have to derive the conditions on $\|\theta_i - \theta_i^*\|$ such that $\|\theta_i - \theta_{i+1}^*\| \leq B$. In addition, we derive the necessary conditions in order to apply these results recursively across the iterations of Algorithm 1.

**Proposition 4.11.** *Let $\theta_i \in \mathcal{B}_{B,\theta_i^*}$ and $|\lambda_i - \lambda_{i+1}| \leq \epsilon$, with $0 \leq \epsilon \leq \frac{B}{\delta}$. If $\|\theta_i - \theta_i^*\| \leq B - \delta\epsilon$, then $\|\theta_i - \theta_{i+1}^*\| \leq B$. Moreover, let $\kappa_d = \sqrt{(1-\alpha\mu)}$ and assume that*

$$\frac{(N-M)}{N} \leq \frac{(1-\kappa_d^k)(1-\kappa_d)B}{2\alpha\nu}\,,$$

*and*

$$\epsilon \leq \frac{1}{\delta}\left((1-\kappa_d^k)B - \frac{(N-M)}{N}\frac{2\alpha\nu}{(1-\kappa_d)}\right)\,.$$

*Then, after applying $k$ iterations of SGD, we obtain that*

$$\|\theta_{i+1} - \theta_{i+1}^*\| \leq B - \delta\epsilon\,.$$

*Proof.* See Section E.1 in the appendix. □

See Figure 9 in the appendix for a graphical representation of the results derived in Proposition 4.11, where the continuous and dashed lines are used to represent the circles of radius $B$ and $B - \delta\epsilon$, respectively.

**Theorem 4.12.** *Consider Algorithm 1 with Stochastic Gradient Descent as solver and let $k > 0$ be the number of iterations, $0 < \alpha \leq \min\left(\frac{1}{2\mu}, \frac{1}{L}\right)$ be the step size and $0 < M \leq N$ be the batch size such that*

$$\frac{(N - M)}{N} \leq \frac{(1 - \kappa_d^k)(1 - \kappa_d)B}{2\alpha\nu},$$

*where $\kappa_d = \sqrt{(1 - \alpha\mu)}$. For $\theta_0 \in \mathcal{B}_{B - \delta\epsilon, \theta_0^*}$ and $r_\theta \in \mathbb{R}$ such that*

$$r_\theta^2 \geq \frac{\alpha C^2}{2\mu}, \tag{11}$$

*then, if $\mathbb{E}\left[\|\theta_i - \theta_i^*\|^2\right] \leq r_\theta^2$ and $|\lambda_i - \lambda_{i+1}| \leq \tilde{\epsilon}$, where $\tilde{\epsilon} := \min\{\bar{\epsilon}, \epsilon\}$ with*

$$\bar{\epsilon} = -\frac{r_\theta}{\delta} + \frac{1}{\delta}\sqrt{\frac{r_\theta^2 - \alpha C^2/2\mu}{(1 - 2\alpha\mu)^k}}, \tag{12}$$

*the following inequality holds*

$$\mathbb{E}\left[\|\theta_{i+1} - \theta_{i+1}^*\|^2\right] \leq r_\theta^2. \tag{13}$$

*Proof.* See Section E.2 in the appendix. □

The results derived in Theorem 4.12 show that the homotopy method used in combination with SGD allows to track in expectation an $r_\theta$-optimal solution across the parametric problems for "small enough" variations of the homotopy parameter, i.e. $\Delta\lambda \leq \tilde{\epsilon}$. Notice that $r_\theta$ can potentially be smaller than $B - \delta\epsilon$ and has to be bigger than the radius of the "noise-dominant" hypersphere centered at the minimizers, i.e. $r_\theta^2 \geq \frac{\alpha C^2}{2\mu}$. In particular, by exploiting the local structure of the parametric problems we derive the maximum allowed variation of the homotopy parameter across the iterations of Algorithm 1. The derived upper bound is inversely proportional to the strong regularity constant $\delta$ and depends on the number of iterations $k$ performed with SGD, such that the more iterations we perform on each parametric problem the more we are allowed to change the homotopy parameter. Finally, notice that these results can be applied recursively across the parametric problems.

## 5 TRANSFERRING OPTIMALITY VIA HOMOTOPY METHODS

In this section we describe a possible application of homotopy methods to solve supervised regression and classification tasks. We address the case where deep neural networks are used as models. We start by introducing the problem framework of supervised learning and then we propose two different homotopy functions for the regression and classification scenarios, respectively.

### 5.1 PROBLEM FORMULATION

Despite the generality of the proposed methodology, in this work we specifically address the supervised learning framework, and, in particular, when the predictive model is constituted by a deep neural network $f(x; \theta)$ parameterized by $\theta \in \mathbb{R}^d$.

In the supervised learning scenario, independently from the type of task $t$, we typically dispose of a training set $\mathcal{D}_t$ consisting of $N$ pairs of examples $(x_j, y_j)$. The goal of the learning process is to find a value of $\theta$ that minimizes an objective function which measures the discrepancy between the outputs produced by the network $\hat{y} = f(x; \theta)$ and the target outputs $y$. In particular, the learning process consists in minimizing the following empirical objective function

$$J(\theta) := \frac{1}{N} \sum_{(x_j, y_j) \in \mathcal{D}_t} \ell(y_j, f(x_j; \theta)), \tag{14}$$

whose non-convexity originates from the high non-convexity of our model $f$.

In the classical setting, $J$ is chosen based on the KL divergence between the target data distribution $Q_{x,y}$, with density $q_{x,y} = q(y|x)q(x)$, and the learned data distribution $P_{x,y}(\theta)$, with density $p_{x,y} = p(y|x;\theta)q(x)$, where $p(y|x;\theta)$ is modeled via a neural network, (Goodfellow et al., 2016). With the appropriate approximations, this leads to the following form for the objective function

$$J(\theta) = \frac{1}{N} \sum_{(x_j,y_j)\in\mathcal{D}_t} q(y|x) \log \frac{q(y|x)}{p(y|x;\theta)} \,. \tag{15}$$

## 5.2 Homotopy Functions Across Data Distributions

Finding a value of $\theta$ that attains a local minimum of the objective function in (14) is often a hard optimization task, given the high dimensionality and non-convexity of the problem. In addition, prior knowledge regarding the localization of the solutions is rarely available. The complexity of minimizing such functions also depends in some non-trivial way on the task distribution $Q_{x,y}$ that is addressed (e.g., Ionescu et al. (2016), Zendel et al. (2017)). For some tasks, convergence to a good approximate solution is achieved after a few epochs, while for other tasks, orders of magnitude more iterations are required to reach the neighborhood of a solution. In this perspective, different heuristics have been recently proposed in the attempt of re-using across different data distributions the prior knowledge gained from approximately solving the learning problem associated with a certain task. The question whether we could exploit easy-to-solve or already-solved tasks to speed up and improve the learning of unsolved hard tasks arises. The method we propose in this paper addresses this question and attempts to do so by using a rigorous and well-established mathematical framework, with the goal of speeding up the learning process in presence of hard-to-solve tasks.

In the perspective of homotopy methods, this goal can be achieved under some assumptions by defining a homotopy transformation between starting and target tasks and by following the procedure described in Algorithm 1. Despite the flexibility and generality of the method, with this work we only focus on homotopy deformations across different task distributions, but similar transformations can be applied in numerous different manners that are also worth exploring, e.g., progressively modifying the architecture of the network or the weights of the objective function terms.

Let $s$ be the source task with training data $\mathcal{D}_s$ of pairs $(x_s, y_s) \sim Q_{x_s,y_s}$ whose good approximate solution $\theta_s^*$ for the minimization of the objective in (14) is available (or cheaply computable), and let $t$ denote the target task with training data $\mathcal{D}_t$ of pairs $(x_t, y_t) \sim Q_{x_t,y_t}$ whose conditional distribution we aim to learn. We propose two different homotopy deformations from task $s$ to task $t$ for regression and classification, respectively.

### 5.2.1 Supervised Regression

In the supervised regression scenario, by modeling the density of the conditional learned distribution as $p(y|x;\theta) = \mathcal{N}\big(y; f(x,\theta), \sigma^2 I\big)$ and using the approximate KL divergence objective function described in (15), we recover the mean squared error as minimization criterion. The proposed homotopy deformation is based on the following equations

$$y_\lambda|x = (1-\lambda)\,y_s|x + \lambda\,y_t|x\,, \tag{16}$$

$$p(y_\lambda|x) = \mathcal{N}(y_\lambda\,;\,f(x;\theta),\sigma^2\,I)\,. \tag{17}$$

Notice that the transformation described in (16) preserves the unimodality of the conditional distribution (see caption of Figures 4a and 4b in the appendix), and, when used in combination with the objective function defined in Equation (15), leads to the minimization w.r.t. $\theta$ of

$$H(\theta, \lambda) := E_{(x,y_\lambda)} \|(1-\lambda)\,(y_s - f(x;\theta)) + \lambda\,(y_t - f(x;\theta))\,\|^2\,. \tag{18}$$

See Figure 6a in the appendix for a graphical representation of this homotopy deformation when applied to gradually transform a one-dimensional sine wave function with a frequency of 1 radian into a one-dimensional sine wave function with a frequency of 137 radians. A downside of this homotopy deformation is that the same support for $x$ is required (the absence of the subscripts $s$ and $t$ on $x$ stands to indicate that the same realization for $x_s$ and $x_t$ has to be considered). Alternatively, it is possible to approximate (16) by using a Gaussian filter (see Figure 6b and Section B in the appendix).

### 5.2.2 SUPERVISED CLASSIFICATION

In the case of supervised classification, by modeling the density of the conditional learned distribution as $p(y|x;\theta) = Multinoulli(y; f(x;\theta))$, and using the approximate KL divergence objective function described in (15), we recover the cross-entropy loss function, (Goodfellow et al., 2016). A possible homotopy deformation for the classification case consists in applying the following transformations

$$x_\lambda = (1 - \lambda)\, x_s + \lambda\, x_t\,, \tag{19}$$

$$y_\lambda|x_\lambda = (1 - \lambda)\, y_s|x_s + \lambda\, y_t|x_t\,, \tag{20}$$

which corresponds to the use of probabilistic labels. See Figure 8 in the appendix for a graphical representation of the proposed homotopy deformation. The corresponding label vector for the deformed image represented in Figure 8b is $y_{0.5} = [0, 0, 0.5, 0, 0, 0.5, 0, 0, 0, 0]$, given that $\lambda = 0.5$ and that the sampled realizations of $x_s$ and $x_t$, represented in Figures 8a and 8c, belong to class 2 and 5, respectively.

## 6 EXPERIMENTAL EVALUATION

In this section, we present some experimental evaluations of homotopy methods when applied to solve supervised regression and classification tasks. As homotopy functions we adopt the ones discussed in Section 5.2. We empirically show that homotopy methods outperform random and warm-start initialization schemes in terms of numerical performance. In particular, when the target task is complex and/or, in the transfer-learning scenario, when the data distributions are significantly different, continuation methods can achieve significant speed-up compared to random and warm-start initializations. We believe that their superior numerical performance relies on the use of homotopy functions that progressively deform the data distribution from an easy-to-solve or already-solved task to the target data distribution. In addition, consistently across all the benchmarks, our homotopy-based method shows faster convergence than random-initialization and faster or comparable convergence than warm-start initialization. When the source task is "similar" to the target one, there is indeed no need to gradually vary the $\lambda$ parameter in Algorithm 1, but it suffices to directly set it to 1. In this extreme case, our homotopy method boils down to warm-start initialization.

### 6.1 REGRESSION

For the supervised regression scenario, the problem we address is how to transfer "optimality knowledge" across two tasks that involve regressing from the input to the output of two sine wave functions with different values of phase $\omega$. Each considered dataset has 10000 samples split across training and testing, where $x$ and $y$ are defined as follows

$$x \sim \mathcal{U}(0, 1)\,, \qquad y = \sin(\omega x) + \varepsilon\,, \quad \varepsilon \sim \mathcal{N}(0, 0.01)\,. \tag{21}$$

The goal is to start with an "easy-to-learn" task, i.e. $\omega \approx 1$ rad, whose optimum is available by performing only few epochs with a first-order optimizer, e.g. SGD, Adam, and progressively transfer the "optimality knowledge" to a more complex task, i.e. $\omega >> 1$ rad, by approximately solving the homotopy problems for increasing values of $\lambda$ as described in Algorithm 1. We set $\omega = 1$ rad for our source task distribution, and study the performance of the proposed approach with homotopy function as described in Equation (16) for different target distributions with $\omega >> 1$ rad. See Figures 5a and 5b in the appendix for a visualization of the source data distribution with $\omega = 1$ rad and the target data distribution when $\omega = 137$ rad, respectively. The regressor is a feedforward neural network with 6 hidden layers of 100 units each and *relu* as activation function. In order to make the experiments more robust with respect to the choice of the step size $\alpha$, we use Adam as optimizer. For the experiments in Figures 1a–1b, Figures 7a–7b in the appendix, and Figure 2a, we set $\alpha = 0.001$, $\gamma = 10$, $k = 200$ and then performed an additional 500 epochs on the final target problem, while for the experiments in Figure 2b, we set $\gamma = 10$, $k = 300$ and performed an additional 600 epochs on the final target problem. In this last scenario we set $\alpha = 0.001$ and then decrease it with a cosine annealing schedule to observe convergence to an optimum. As shown in Figures 1a–1b, Figures 7a–7b in the appendix, and Figures 2a and 2b, the homotopy method leads to faster convergence than the considered baselines by preserving the vicinity to an optimal solution for problems $H(\theta, \lambda)$ across the different $\lambda$ values. In particular, we achieve a training loss up to two orders of magnitude better than the considered baselines.

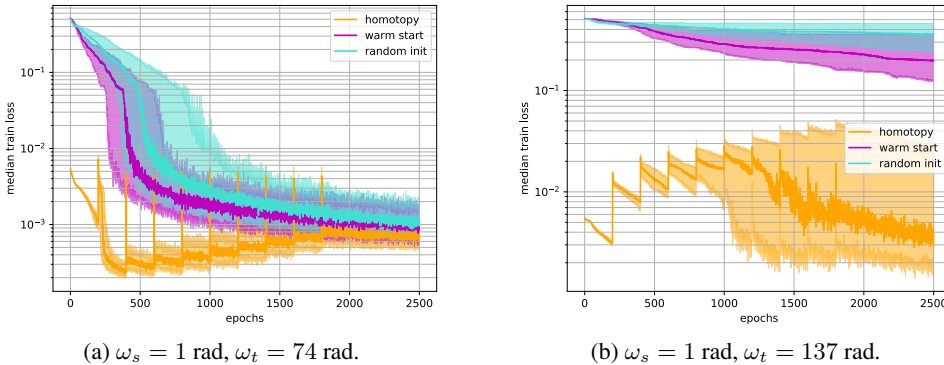

(a) $\omega_s = 1$ rad, $\omega_t = 74$ rad.

(b) $\omega_s = 1$ rad, $\omega_t = 137$ rad.

Figure 1: Median train loss across 100 runs versus epochs for sine wave regression tasks with different frequency values. The shaded areas represent the 25th and 75th percentiles. See Section F.1 in the appendix for an evaluation of the test performance.

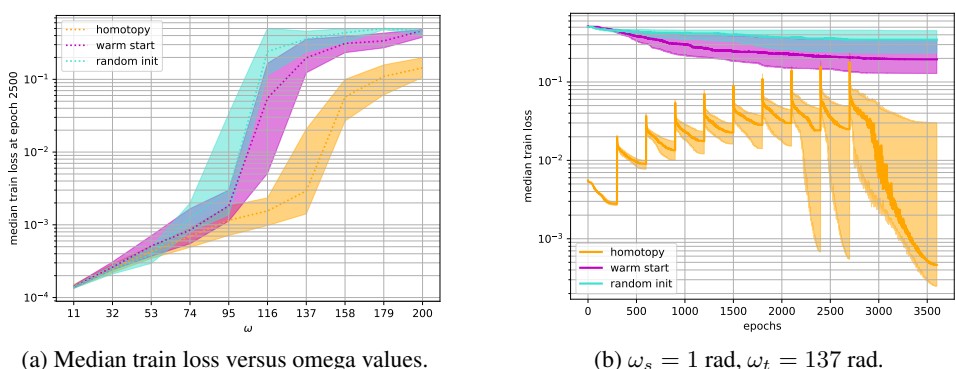

(a) Median train loss versus omega values.

(b) $\omega_s = 1$ rad, $\omega_t = 137$ rad.

Figure 2: Comparison of homotopy method, warm start and random initialization on sine wave regression tasks. The shaded areas represent the 25th and 75th percentiles. On the left, the median train loss achieved by the considered methods after 2500 epochs across 100 runs versus different omega values for the target task is plotted. For the homotopy method and warm-start initialization, $\omega_s = 1$ rad is used. On the right, the median train loss across 100 runs versus epochs for target task with $\omega = 137$ rad is plotted. With respect to Figure 1b, in Figure 2b a cosine decay schedule is used for the learning rate, and more epochs are performed to better observe the convergence properties of the different methods.

## 6.2 CLASSIFICATION

For the supervised classification scenario, we first apply the continuation method with the homotopy deformation described in Equations (19) and (20) in order to transfer optimality from the MNIST task, a notoriously "easy-to-learn" task for neural networks, to the FashionMNIST task. Since the two datasets have the same input dimensionality and the same number of classes, no additional pre-processing of the data is required. As network architecture, we use a VGG-type network, (Simonyan & Zisserman, 2015), and Adam as optimizer with a step size of $\alpha = 0.001$.

Secondly, we consider CIFAR-10 as target data distribution. Differently from the previous scenario, padding of the MNIST samples is required in order to apply Equation (19). The MNIST samples are also replicated across three channels. Also in this case we adopt a VGG-type network, (Simonyan & Zisserman, 2015), and Adam as optimizer with a step size of $\alpha = 0.0001$.

As shown in Figures 3a and 3b, in both benchmarks the homotopy method leads to faster convergence than random initialization. While in the second benchmark our method reaches a lower value of training loss in fewer epochs than warm-start, in the MNIST-to-FashionMNIST case the performance is comparable to using warm-start initialization. A possible interpretation is that, when the source and target task distributions are "too similar", as we hypothesize in the MNIST-

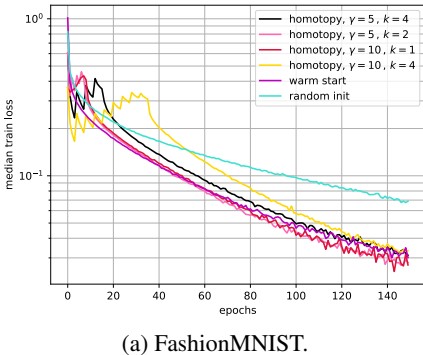
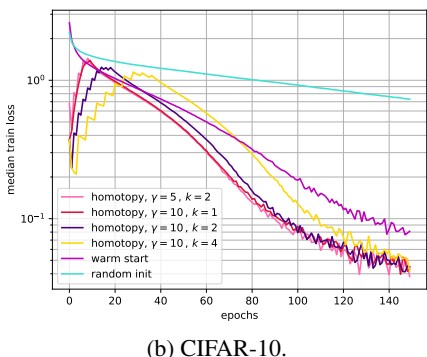

(a) FashionMNIST.          (b) CIFAR-10.

Figure 3: Median train loss across 10 runs versus epochs for different target task distributions. In both cases, the source task is the classification of the MNIST dataset. See Section F.2 in the appendix for an evaluation of the test performance.

to-FashionMNIST scenario, then there is no need for homotopy deformations to be applied, i.e. $0 < \lambda < 1$, but we can directly apply $\lambda = 1$ in our scheme, which corresponds to simply using warm-start initialization.

## 7 CONCLUSIONS

In this paper we propose a new methodology based on homotopy methods in order to transfer knowledge across different task distributions. In particular, our homotopy-based method allows one to exploit easy-to-solve or already-solved learning problems to solve new and complex tasks, by approximately and sequentially solving a sequence of optimization problems where the task distribution is gradually deformed from the source to the target one. We conduct a theoretical analysis of a general homotopy method in a simplified setting, and then we test our method on some popular deep learning benchmarks, where it shows superior numerical performance compared to random and warm-start initialization schemes. The proposed framework, in its limiting case, corresponds to the widely used fine-tuning heuristic, allowing for a new and more rigorous interpretation of the latter. Finally, the generality of homotopy methods also opens many novel and promising research directions in fundamental fields for deep learning, such as stochastic non-convex optimization and transfer-learning.

### ACKNOWLEDGMENTS

This work has partly been supported by the European Research Council (ERC) under the European Union's Horizon 2020 research and innovation programme under grant no. 716721 as well as by the German Federal Ministry for Economic Affairs and Energy (BMWi) via DyConPV (0324166B), and by DFG via Research Unit FOR 2401. In addition, Q. Tran-Dinh has partly been supported by the National Science Foundation (NSF), grant. no. 1619884. The authors thank Stefan Falkner for his helpful suggestions and comments.

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

## A    PROPERTIES OF HOMOTOPIC FUNCTIONS

Among the numerous properties of homotopic functions, we recall the following ones

**Proposition A.1.** *Suppose that there exists a homotopy $H : Z \times [0,1] \to Y$ from $g$ to $f$, i.e. $g \simeq f$. Then*

- $g \simeq g$                                        *(reflexive property)*
- $g \simeq f \implies f \simeq g$                  *(symmetric property)*
- $g \simeq f$  *and*  $f \simeq h \implies g \simeq h$      *(transitive property)*

*Proof.* See proof of Theorem 1.5 in (Suciu, 2016).          □

**Proposition A.2.** *Let $g, g' : Z \to Y$ and $f, f' : Y \to W$ be continuous maps, and let $f \circ g$, $f' \circ g' : Z \to W$ be the respective composite maps. If $g \simeq g'$ and $f \simeq f'$, then $f \circ g \simeq f' \circ g'$.*

*Proof.* See proof of Proposition 1.7 in (Suciu, 2016).          □

## B    APPROXIMATION VIA GAUSSIAN FILTER

For the supervised regression scenario, we propose the following homotopy deformation

$$y_\lambda | x = \lambda\, y_s | x + (1 - \lambda)\, y_t | x. \tag{22}$$

A downside of this homotopy function is that the same support for $x$ is required (the absence of the subscripts $s$ and $t$ on $x$ stands to indicate that the same realization for $x_s$ and $x_t$ has to be considered). Alternatively, it is possible to approximate Equation (22) by using a Gaussian filter, as depicted in Figure 6b.

In particular, having sampled one realization $z$ of the pair $(x_s, y_s)$ from the training set $\mathcal{D}_s$, $0 < M_{GF} \leq N$ realizations of the pair $(x_t, y_t)$ are sampled from $\mathcal{D}_t$. Each $y_{t,j}$ realization is then weighted based on the vicinity of $x_{t,j}$ to the sampled $x_{s,z}$ realization. This leads to the following approximation of the $z$ realization of $y_\lambda$

$$y_{\lambda,z} = (1 - \lambda)\, y_{s,z} + \frac{\lambda}{M_{GF}} \sum_{j=1}^{M_{GF}} w_j\, y_{t,j}, \tag{23}$$

$$w_j = \frac{1}{\sqrt{2\pi\xi^2}} \exp\left(-\frac{||x_{s,z} - x_{t,j}||^2}{2\xi^2}\right), \tag{24}$$

where $\xi > 0$ is the standard deviation of the Gaussian filter.

## C    ADDITIONAL FIGURES

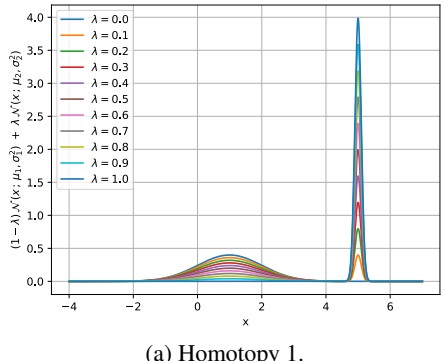
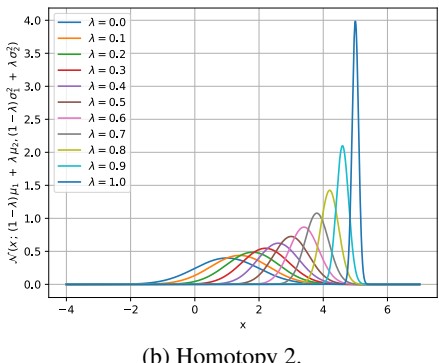

(a) Homotopy 1.                              (b) Homotopy 2.

Figure 4: Two different homotopy deformations between the probability density functions of two one-dimensional Gaussian distributions with mean and standard deviation given by $\mu_1 = 1$, $\sigma_1 = 1$ and $\mu_2 = 5$, $\sigma_2 = 0.1$, respectively. The homotopy represented in Figure 4a results in a mixture of Gaussian distributions, with mixture coefficient given by the homotopy parameter $\lambda$. In Figure 4b the deformation concerns instead the parameters $\mu$ and $\sigma$ of the original distributions. Preserving unimodality is a desirable property when the homotopy function is used in combination with a continuation method since, as shown in Figure 4b, the location of the optimum moves together with the function deformation, allowing the optimizer to track it and gradually reach the optimum of the final target task. On the contrary, deforming the function as shown in Figure 4a does not lead to a gradual shift of the optimal solutions. Consequently, approximately and sequentially solving the problems corresponding to intermediate values of the homotopy parameter $\lambda$, i.e. $0 < \lambda < 1$, will not allow the homotopy method to gradually approach the desired final optimal solution.

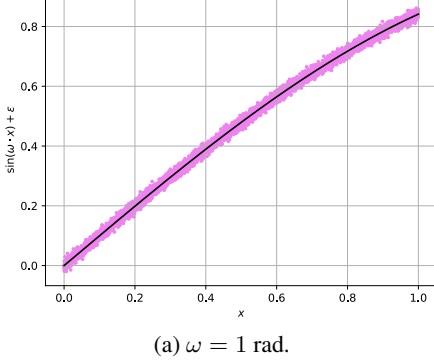
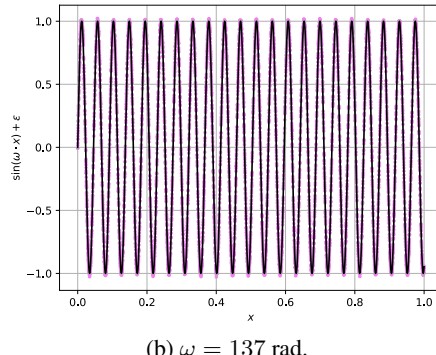

(a) $\omega = 1$ rad.                           (b) $\omega = 137$ rad.

Figure 5: Graphical representations of the source (left) and target with $\omega = 137$ rad (right) data distributions used for the sine-wave regression evaluation.

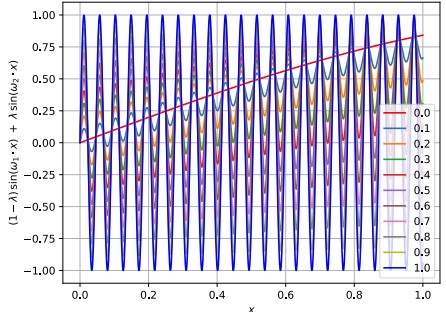

(a) Homotopy transformation described in Equation (16).

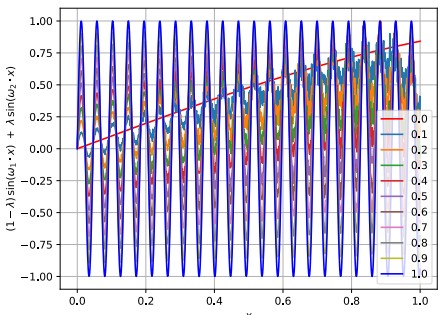

(b) Approximation of the homotopy transformation in Equation (16) (also Equation (22)) with a Gaussian filter as described in Equations (23) and (24).

Figure 6: Graphical representation of the proposed homotopy transformation for the supervised regression scenario when applied to progressively deform a sine wave function with frequency of 1 radian into a sine wave function with frequency of 137 radians for different values of homotopy parameter.

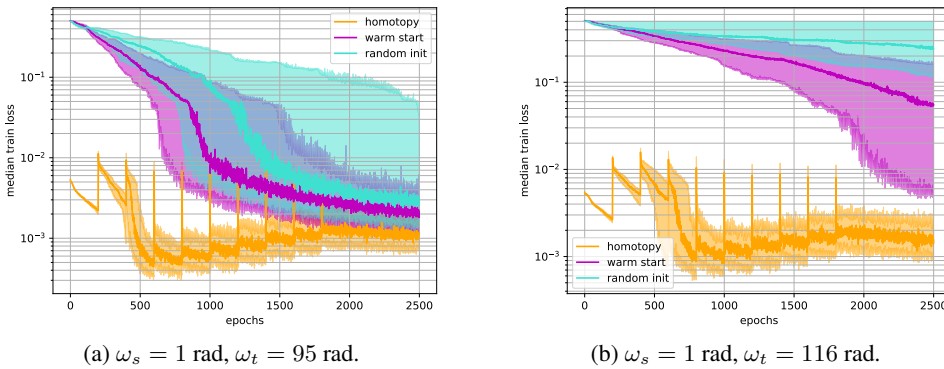

(a) $\omega_s = 1$ rad, $\omega_t = 95$ rad.

(b) $\omega_s = 1$ rad, $\omega_t = 116$ rad.

Figure 7: Median train loss across 100 runs versus epochs for sine wave regression tasks with different omega values.

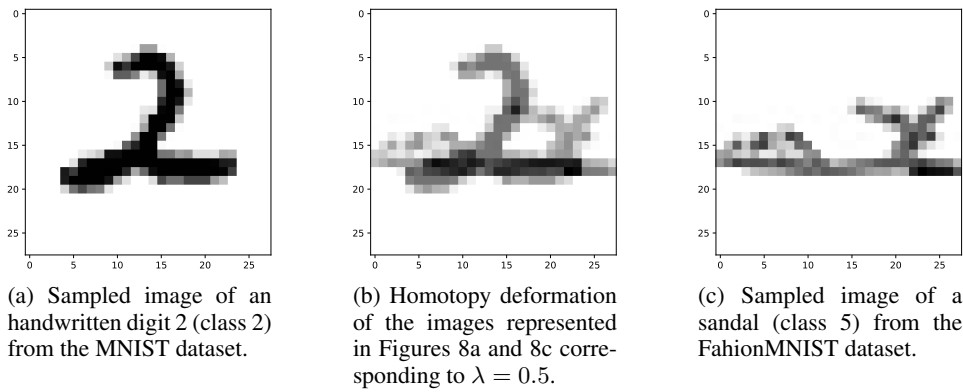

(a) Sampled image of an handwritten digit 2 (class 2) from the MNIST dataset.

(b) Homotopy deformation of the images represented in Figures 8a and 8c corresponding to $\lambda = 0.5$.

(c) Sampled image of a sandal (class 5) from the FahionMNIST dataset.

Figure 8: Graphical representation of the homotopy transformation from $x_s$ to $x_t$ as described in Equation (19) for two sampled images from the MNIST and FashionMNIST datasets.

## D   LOCAL ERROR BOUNDS FOR SGD ITERATES

Before proving local error bounds for SGD iterates in the considered framework, given the local nature of our assumptions, we need to demonstrate two important facts, on which the proof relies. In particular, we need to show:

- local linear contraction of Gradient Descent (GD) iterates, and that
- starting in a hypersphere of radius $B$ around a minimizer and given a "big enough" batch size, the next SGD iterate is also contained in this region for all possible realizations of the gradient estimate.

Considering problem (4) with fixed parameter $\lambda_i$, in the following subsections we will refer to $\theta^* = \theta_i^*$, $\theta_k = \theta_{i,k}$ and $g_k = g(\theta_k, \lambda_i)$, where we drop the subscript $_i$ and the explicit dependence on $\lambda_i$ in order to simplify the notation. The analysis holds for all fixed parameters $\lambda_i$.

### D.1   LOCAL LINEAR CONTRACTION OF GD ITERATES

Let us use GD to solve the following optimization problem

$$\theta^* \in \arg\min_\theta H(\theta, \lambda_i) \,,$$

where the objective function $H$ fulfills Assumptions 4.2 and 4.3.

We now derive error bounds on the iterates of GD

$$\theta_{k+1} = \theta_k - \alpha \nabla_\theta H(\theta_k, \lambda_i) \,,$$

where $\theta_k \in \mathcal{B}_{B,\theta^*}$ and $0 < \alpha \leq \frac{1}{L}$ is the step size.

We start by applying the definition of GD iterates and then we exploit the introduced assumptions

$$
\begin{aligned}
\|\theta_{k+1} - \theta^*\|^2 &= \|\theta_k - \alpha\nabla_\theta H(\theta_k, \lambda_i) - \theta^*\|^2 \\
&= \|\theta_k - \theta^*\|^2 - 2\alpha\nabla_\theta H(\theta_k, \lambda_i)^T(\theta_k - \theta^*) + \alpha^2\|\nabla_\theta H(\theta_k, \lambda_i)\|^2 \\
&\overset{\text{strong convexity}}{\leq} (1 - \alpha\mu)\|\theta_k - \theta^*\|^2 - 2\alpha(H(\theta_k, \lambda_i) - H(\theta^*, \lambda_i)) + \alpha^2\|\nabla_\theta H(\theta_k, \lambda_i)\|^2 \\
&\overset{\text{corollary 4.2.1}}{\leq} (1 - \alpha\mu)\|\theta_k - \theta^*\|^2 - 2\alpha(1 - \alpha L)(H(\theta_k, \lambda_i) - H(\theta^*, \lambda_i)) \,.
\end{aligned}
$$

Since $H(\theta_k, \lambda_i) - H(\theta^*, \lambda_i) \geq 0$ and $-2\alpha(1 - \alpha L) \leq 0$ when $0 < \alpha \leq \frac{1}{L}$, we can safely drop the second term and obtain the final result

$$\|\theta_{k+1} - \theta^*\|^2 \leq (1 - \alpha\mu)\|\theta_k - \theta^*\|^2 .$$

See also Theorem 2.3 in (Gower, 2018) for a derivation where Assumptions 4.2 and 4.3 are required to hold globally.

### D.2   REALIZATION OF THE SGD ITERATES IN THE STRONG CONVEXITY AND L-SMOOTHNESS REGION AROUND A MINIMIZER

We address the following optimization problem

$$\theta^* \in \arg\min_\theta \underbrace{\frac{1}{N}\sum_{j=1}^N \ell_j(\theta, \lambda_i)}_{:=H(\theta,\lambda_i)} \,,$$

where $H$ fulfills Assumptions 4.2– 4.4.

As proved in Section D.1, under Assumptions 4.2 and 4.3, whenever $\theta_0 \in \mathcal{B}_{B,\theta^*}$ and $0 < \alpha \leq \frac{1}{L}$, deterministic gradient descent iterates converge linearly with contraction rate $\kappa_d := \sqrt{(1 - \alpha\mu)}$.

In particular, the following inequality holds

$$\|\theta_{k+1}^D - \theta^*\| \leq \kappa_d \cdot \|\theta_k - \theta^*\| \,,$$

for any $\theta_k$ such that $\|\theta_k - \theta^*\| \leq B$, and superscript $^D$ denotes iterates obtained by applying the full gradient $\nabla H_k := \nabla H(\theta_k, \lambda_i)$

$$\theta_{k+1}^D = \theta_k - \alpha \nabla H_k \,.$$

Let $\theta_{k+1}$ denote the iterate obtained by applying one iteration of stochastic gradient descent

$$\theta_{k+1} = \theta_k - \alpha g_k \,,$$

where $g_k := \frac{1}{M} \sum_{j \in \mathcal{M}} \nabla \ell_j(\theta_k, \lambda_i)$ and $\mathcal{M}$ is a set of $0 < M \leq N$ indexes randomly sampled from $\mathcal{N} = \{1, \ldots, N\}$.

Given any realization of $\theta_k$ s.t. $\|\theta_k - \theta^*\| \leq B$ and any realization of $g_k$, by exploiting Assumption 4.4 and the results derived in Section D.1, we have that

$$
\begin{aligned}
\|\theta_{k+1} - \theta^*\| &= \|\theta_k - \alpha g_k - \theta^*\| \\
&= \|\theta_k - \alpha \nabla H_k + \alpha \nabla H_k - \alpha g_k - \theta^*\| \\
&\leq \|\theta_k - \alpha \nabla H_k - \theta^*\| + \alpha \|\nabla H_k - g_k\| \\
&= \|\theta_k - \alpha \nabla H_k - \theta^*\| + \alpha \left\| \frac{1}{N} \sum_{j \in \mathcal{N} \backslash \mathcal{M}} \nabla \ell_j + \frac{1}{N} \sum_{j \in \mathcal{M}} \nabla \ell_j - \frac{1}{M} \sum_{j \in \mathcal{M}} \nabla \ell_j \right\| \\
&= \|\theta_k - \alpha \nabla H_k - \theta^*\| + \alpha \left\| \frac{1}{N} \sum_{j \in \mathcal{N} \backslash \mathcal{M}} \nabla \ell_j + \frac{M - N}{NM} \sum_{j \in \mathcal{M}} \nabla \ell_j \right\| \\
&\leq \|\theta_k - \alpha \nabla H_k - \theta^*\| + \alpha \left( \frac{1}{N} \sum_{j \in \mathcal{N} \backslash \mathcal{M}} \|\nabla \ell_j\| + \frac{N - M}{NM} \sum_{j \in \mathcal{M}} \|\nabla \ell_j\| \right) \\
&\leq \|\theta_{k+1}^D - \theta^*\| + 2\alpha \frac{(N - M)}{N} \nu \\
&\leq \kappa_d \|\theta_k - \theta^*\| + 2\alpha \frac{(N - M)}{N} \nu \,.
\end{aligned}
\tag{25}
$$

Since we have assumed that the current realization of $\theta_k$ lies in the hypersphere of radius $B$ around the optimal solution $\theta^*$, by solving for $\frac{N-M}{N}$ the following inequality

$$\kappa_d B + 2\alpha \frac{(N - M)}{N} \nu \leq B \,,$$

we obtain that, whenever $\frac{(N-M)}{N} \leq \frac{(1-\kappa_d)}{2\alpha\nu} B$, the realization of $\theta_{k+1}$ will also lie in this region.

These derivations show that when the realization of the current iterate $\theta_k$ lies in the hypersphere of radius $B$ around the minimizer $\theta^*$, and $\frac{(N-M)}{N} \leq \frac{(1-\kappa_d)}{2\alpha\nu} B$, then the next iterate $\theta_{k+1}$ will also lie in this region. Consequently, in our scenario, if we assume that the initial point $\theta_0$ lies in the hypersphere of radius $B$ around the minimizer $\theta^*$, then, by applying the derivations recursively, we can show that the iterates will remain in this local region around the minimizer where strong convexity and smoothness hold.

### D.3 PROOF OF PROPOSITION 4.9

Let us use SGD to solve the following optimization problem

$$\theta^* \in \arg\min_\theta H(\theta, \lambda_i) \,,$$

where the objective function $H$ fulfills Assumptions 4.2– 4.4. We now derive error bounds for the iterates of SGD

$$\theta_{k+1} = \theta_k - \alpha g_k \,,$$

where $g_k$ is the unbiased estimate of $\nabla H_k$ defined in the previous section and fulfills Assumption 4.5, $\theta_k \in \mathcal{B}_{B,\theta^*}$, $0 < \alpha \leq \min\left(\frac{1}{2\mu}, \frac{1}{L}\right)$ is the step size and the batch size is set to a value $M$ such that $\frac{(N-M)}{N} \leq \frac{(1-\kappa_d)}{2\alpha\nu} B$.

We start by applying the definition of SGD iterates

$$\|\theta_{k+1} - \theta^*\|^2 \overset{\text{SGD iterate}}{=} \|\theta_k - \alpha g_k - \theta^*\|^2$$
$$= \|\theta_k - \theta^*\|^2 - 2\alpha g_k^T(\theta_k - \theta^*) + \alpha^2 \|g_k\|^2 \, .$$

We now take the expectation w.r.t. $\theta_0, g_0, \ldots, g_{k-1}, g_k$ and, considering Assumptions 4.2- 4.5, we obtain the following series of inequalities

$$\mathbb{E}_{\theta_0, g_0, \ldots, g_{k-1}, g_k} \left[ \|\theta_{k+1} - \theta^*\|^2 \right] = \mathbb{E}_{\theta_0, g_0, \ldots, g_{k-1}, g_k} \left[ \|\theta_k - \theta^*\|^2 - 2\alpha g_k^T(\theta_k - \theta^*) \right.$$
$$\left. + \alpha^2 \|g_k\|^2 \right]$$

$$\overset{\text{law of iterated expectations}}{=} \mathbb{E}_{\theta_0, g_0, \ldots, g_{k-1}} \left[ \mathbb{E}_{g_k} \left[ \|\theta_k - \theta^*\|^2 \right.\right.$$
$$\left.\left. - 2\alpha g_k^T(\theta_k - \theta^*) + \alpha^2 \|g_k\|^2 \, | \, \theta_0, g_0, \ldots, g_{k-1} \right] \right]$$

$$\overset{\text{unbiased } g_k \text{ +bounded "variance"}}{\leq} \mathbb{E}_{\theta_0, g_0, \ldots, g_{k-1}} \left[ \|\theta_k - \theta^*\|^2 \right.$$
$$\left. - 2\alpha \nabla H_k^T(\theta_k - \theta^*) \right] + \alpha^2 C^2$$

$$\overset{\text{strong convexity}}{\leq} (1 - 2\alpha\mu) \cdot \mathbb{E}_{\theta_0, g_0, \ldots, g_{k-1}} \left[ \|\theta_k - \theta^*\|^2 \right] + \alpha^2 C^2 \, .$$

By applying this result recursively, we derive the following bound on the error for the SGD iterates

$$\mathbb{E}_{\theta_0, g_0, \ldots, g_{k-1}, g_k} \left[ \|\theta_{k+1} - \theta^*\|^2 \right] \leq (1 - 2\alpha\mu)^{k+1} \cdot \mathbb{E}_{\theta_0} \left[ \|\theta_0 - \theta^*\|^2 \right] + \frac{\alpha C^2}{2\mu} \, .$$

See also Section 3 in (Schmidt, 2014) for a derivation where Assumptions 4.2 and 4.3 are required to hold globally.

# E    MAIN THEORETICAL CONTRIBUTIONS

## E.1    PROOF OF PROPOSITION 4.11

**Proposition E.1.** *Let $\theta_i \in \mathcal{B}_{B, \theta_i^*}$ and $|\lambda_i - \lambda_{i+1}| \leq \epsilon$, with $0 \leq \epsilon \leq \frac{B}{\delta}$. If $\|\theta_i - \theta_i^*\| \leq B - \delta\epsilon$, then $\|\theta_i - \theta_{i+1}^*\| \leq B$. Moreover, let $\kappa_d = \sqrt{(1 - \alpha\mu)}$ and assume that*

$$\frac{(N - M)}{N} \leq \frac{(1 - \kappa_d^k)(1 - \kappa_d)B}{2\alpha\nu} \, ,$$

*and*

$$\epsilon \leq \frac{1}{\delta} \left( (1 - \kappa_d^k)B - \frac{(N - M)}{N} \frac{2\alpha\nu}{(1 - \kappa_d)} \right) \, .$$

*Then, after applying $k$ iterations of SGD, we obtain that*

$$\|\theta_{i+1} - \theta_{i+1}^*\| \leq B - \delta\epsilon \, .$$

*Proof.*

$$\|\theta_i - \theta_{i+1}^*\| = \|\theta_i - \theta_i^* + \theta_i^* - \theta_{i+1}^*\|$$

$$\overset{\text{Triangle Ineq.}}{\leq} \|\theta_i - \theta_i^*\| + \|\theta_i^* - \theta_{i+1}^*\|$$

$$\overset{\text{Assumption 4.7}}{\leq} \|\theta_i - \theta_i^*\| + \delta|\lambda_i - \lambda_{i+1}| \, .$$

Finally, using the fact that $|\lambda_i - \lambda_{i+1}| \leq \epsilon$, it follows that, if $\|\theta_i - \theta_i^*\| \leq B - \delta\epsilon$ with $0 \leq \epsilon \leq \frac{B}{\delta}$, then $\|\theta_i - \theta_{i+1}^*\| \leq B$.

We now derive the conditions on $\epsilon$ such that $\|\theta_{i+1} - \theta_{i+1}^*\| \leq B - \delta\epsilon$. By applying recursively the results derived in Section D.2 (25), we obtain that

$$\|\theta_{i+1} - \theta_{i+1}^*\| \leq \kappa_d^k \|\theta_i - \theta_{i+1}^*\| + 2\alpha \frac{(N - M)}{N} \nu \sum_{i=0}^{k-1} \kappa_d^i \, .$$

By using the limit of the geometric series, we have that

$$\|\theta_{i+1} - \theta_{i+1}^*\| \leq \kappa_d^k \|\theta_i - \theta_{i+1}^*\| + \frac{(N-M)}{N} \frac{2\alpha\nu}{(1-\kappa_d)} \,.$$

Finally, by considering that $\|\theta_i - \theta_{i+1}^*\| \leq B$ and by solving in $\epsilon$ the following inequality

$$\kappa_d^k B + \frac{(N-M)}{N} \frac{2\alpha\nu}{(1-\kappa_d)} \leq B - \delta\epsilon \,,$$

we obtain the following upper bound on $\epsilon$

$$\epsilon \leq \frac{1}{\delta} \left( (1-\kappa_d^k)B - \frac{(N-M)}{N} \frac{2\alpha\nu}{(1-\kappa_d)} \right) \,,$$

from which also the extra condition on the batch size

$$\frac{(N-M)}{N} \leq \frac{(1-\kappa_d^k)(1-\kappa_d)B}{2\alpha\nu} \,.$$

$\square$

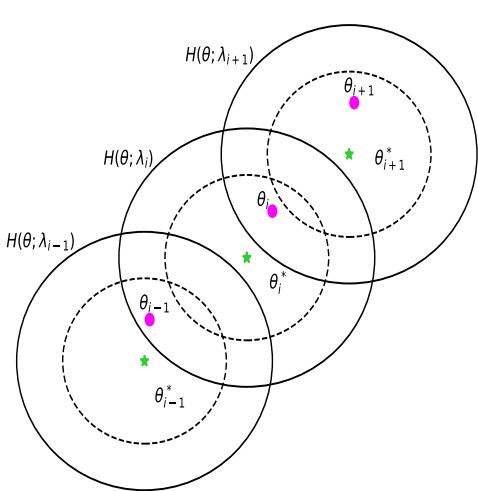

Figure 9: Graphical representation of the results derived in Proposition 4.11. The continuous and dashed lines are used to represent the circles of radius $B$ and $B - \delta\epsilon$ around the optimal solutions, respectively.

### E.2 PROOF OF THEOREM 4.12

**Theorem E.2.** *Consider Algorithm 1 with Stochastic Gradient Descent as solver and let $k > 0$ be the number of iterations, $0 < \alpha \leq \min\left(\frac{1}{2\mu}, \frac{1}{L}\right)$ be the step size and $0 < M \leq N$ be the batch size such that*

$$\frac{(N-M)}{N} \leq \frac{(1-\kappa_d^k)(1-\kappa_d)B}{2\alpha\nu} \,,$$

*where $\kappa_d = \sqrt{(1-\alpha\mu)}$. For $\theta_0 \in \mathcal{B}_{B-\delta\epsilon,\theta_0^*}$ and $r_\theta \in \mathbb{R}$ such that*

$$r_\theta^2 \geq \frac{\alpha C^2}{2\mu} \,, \tag{26}$$

*then, if $\mathbb{E}\left[\|\theta_i - \theta_i^*\|^2\right] \leq r_\theta^2$ and $|\lambda_i - \lambda_{i+1}| \leq \tilde{\epsilon}$, where $\tilde{\epsilon} := \min\{\bar{\epsilon}, \epsilon\}$ with*

$$\bar{\epsilon} = -\frac{r_\theta}{\delta} + \frac{1}{\delta}\sqrt{\frac{r_\theta^2 - \alpha C^2/2\mu}{(1-2\alpha\mu)^k}} \,, \tag{27}$$

*the following inequality holds*

$$\mathbb{E}\left[\|\theta_{i+1} - \theta_{i+1}^*\|^2\right] \leq r_\theta^2 \,. \tag{28}$$

*Proof.*

$$\mathbb{E}\left[\|\theta_{i+1} - \theta_{i+1}^*\|^2\right] \overset{\text{Ineq. 10}}{\leq} (1 - 2\alpha\mu)^k \mathbb{E}\left[\|\theta_i - \theta_{i+1}^*\|^2\right] + \frac{\alpha C^2}{2\mu}$$

$$= (1 - 2\alpha\mu)^k \mathbb{E}\left[\|\theta_i - \theta_i^* + \theta_i^* - \theta_{i+1}^*\|^2\right] + \frac{\alpha C^2}{2\mu}$$

$$\overset{\text{Triangle Ineq.}}{\leq} (1 - 2\alpha\mu)^k \mathbb{E}\left[\left(\|\theta_i - \theta_i^*\| + \|\theta_i^* - \theta_{i+1}^*\|\right)^2\right] + \frac{\alpha C^2}{2\mu}$$

$$= (1 - 2\alpha\mu)^k \mathbb{E}\left[\left(\|\theta_i - \theta_i^*\|^2 + \|\theta_i^* - \theta_{i+1}^*\|^2\right.\right.$$
$$\left.\left. + 2\|\theta_i - \theta_i^*\|\|\theta_i^* - \theta_{i+1}^*\|\right)\right] + \frac{\alpha C^2}{2\mu}$$

$$\overset{\text{Assumption 4.7}}{\leq} (1 - 2\alpha\mu)^k \mathbb{E}\left[\left(\|\theta_i - \theta_i^*\|^2 + \delta^2|\lambda_i - \lambda_{i+1}|^2\right.\right.$$
$$\left.\left. + 2\delta\|\theta_i - \theta_i^*\||\lambda_i - \lambda_{i+1}|\right)\right] + \frac{\alpha C^2}{2\mu}$$

$$\leq (1 - 2\alpha\mu)^k \left(\delta^2\tilde{\epsilon}^2 + 2\delta r_\theta\tilde{\epsilon} + r_\theta^2\right) + \frac{\alpha C^2}{2\mu} \,.$$

We now solve in $\tilde{\epsilon}$ the following second degree inequality

$$(1 - 2\alpha\mu)^k \left(\delta^2\tilde{\epsilon}^2 + 2\delta r_\theta\tilde{\epsilon} + r_\theta^2\right) + \frac{\alpha C^2}{2\mu} \leq r_\theta^2 \,. \tag{29}$$

The inequality (29) admits solutions if and only if $r_\theta^2 \geq \frac{\alpha C^2}{2\mu}$. In particular, inequality (29) holds $\forall \tilde{\epsilon} \in [0, \bar{\epsilon}]$, where $\bar{\epsilon} = -\frac{r_\theta}{\delta} + \frac{1}{\delta}\sqrt{\frac{r_\theta^2 - \alpha C^2/2\mu}{(1 - 2\alpha\mu)^k}}$. $\qquad\square$

## F    EXPERIMENTAL EVALUATION: TEST PERFORMANCES

### F.1    REGRESSION

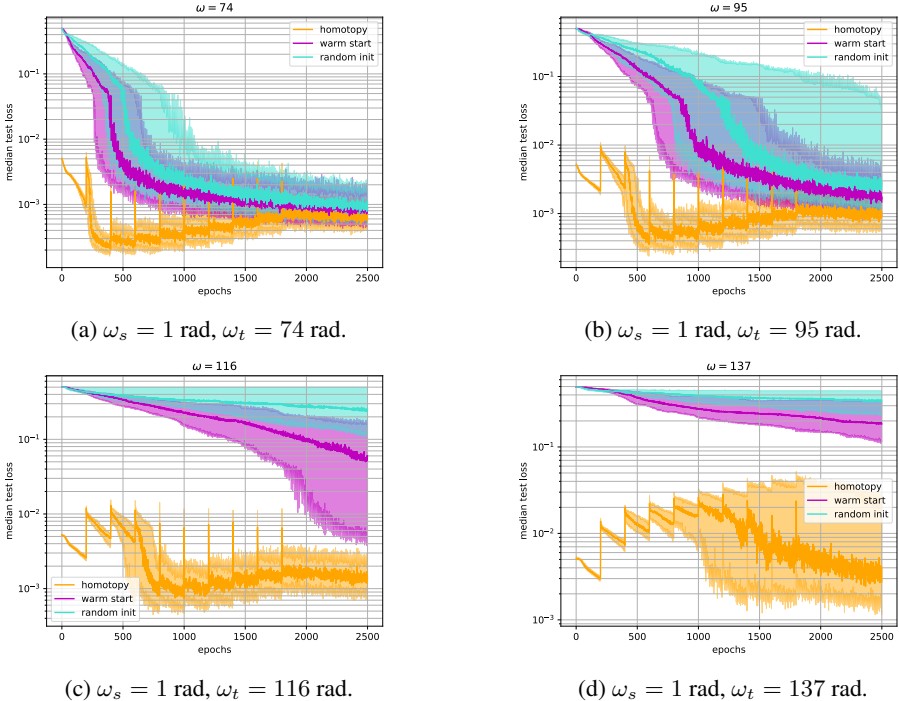

(a) $\omega_s = 1$ rad, $\omega_t = 74$ rad.

(b) $\omega_s = 1$ rad, $\omega_t = 95$ rad.

(c) $\omega_s = 1$ rad, $\omega_t = 116$ rad.

(d) $\omega_s = 1$ rad, $\omega_t = 137$ rad.

Figure 10: Median test loss across 100 runs versus epochs for target tasks with different $\omega$ values. The shaded areas represent the 25th and 75th percentiles. For warm-start initialization and homotopy method, $\omega_s = 1$ rad is used for the source task.

### F.2    CLASSIFICATION

| Method | Final Mean Test Accuracy | Best Mean Test Accuracy |
|---|---|---|
| homotopy $\gamma = 5$, $k = 2$ | $0.89 \pm 0.003$ | $0.91 \pm 0.002$ |
| homotopy $\gamma = 5$, $k = 4$ | $0.89 \pm 0.002$ | $0.91 \pm 0.003$ |
| homotopy $\gamma = 10$, $k = 1$ | $0.89 \pm 0.004$ | $0.91 \pm 0.001$ |
| homotopy $\gamma = 10$, $k = 4$ | $0.90 \pm 0.002$ | $0.91 \pm 0.003$ |
| warm start | $0.89 \pm 0.003$ | $0.90 \pm 0.002$ |
| random init | $0.89 \pm 0.004$ | $0.90 \pm 0.003$ |

Table 1: MNIST-FashionMNIST

| Method | Final Mean Test Accuracy | Best Mean Test Accuracy |
|---|---|---|
| homotopy $\gamma = 5$, $k = 2$ | $0.55 \pm 0.004$ | $0.59 \pm 0.003$ |
| homotopy $\gamma = 10$, $k = 1$ | $0.55 \pm 0.005$ | $0.60 \pm 0.002$ |
| homotopy $\gamma = 10$, $k = 2$ | $0.56 \pm 0.003$ | $0.60 \pm 0.003$ |
| homotopy $\gamma = 10$, $k = 4$ | $0.56 \pm 0.005$ | $0.61 \pm 0.004$ |
| warm start | $0.54 \pm 0.006$ | $0.59 \pm 0.005$ |
| random init | $0.64 \pm 0.02$ | $0.64 \pm 0.02$ |

Table 2: MNIST-CIFAR-10

