# OpenReview forum: "Transferring Optimality Across Data Distributions via Homotopy Methods"
_ICLR.cc/2020/Conference — Accept (Poster)_

### Official Review · AnonReviewer1 · 2019-10-22
**Official Blind Review #1**

**Rating:** 3

**Review:**

Authors propose a very general framework of Homotopy to the deep learning set up and explores a few relevant theoretical issues.

Though the proposed idea is interesting, the depth and breadth of authors' presentation are simply lacking. The entire paper lacks focus and I suggest authors consider focusing on 1-2 well thought-out ideas. There are many 3-4 line long sentences that are hard to decipher. Please also consider making the presentation more accessible.

Overall, this paper does not meet the bar for ICLR.

**Experience Assessment:**

I do not know much about this area.

**Review Assessment: Checking Correctness Of Derivations And Theory:**

I assessed the sensibility of the derivations and theory.

**Review Assessment: Checking Correctness Of Experiments:**

I assessed the sensibility of the experiments.

**Review Assessment: Thoroughness In Paper Reading:**

I made a quick assessment of this paper.

---

> ### Author Response · Authors · 2019-11-08
> **updating abstract+intro+related-work**
>
> Thank you for taking the time to review the paper and for your comments. We are glad to hear that you find the proposed idea interesting.
>
> However, we respectfully disagree with your assessment that our paper lacks focus. We introduce *a single novel idea*, which we substantiate with both theoretical and empirical results: to use the homotopy method to track the optimum of a neural network from one data distribution to another. In particular, after introduction and related work, we derive theoretical results for combining SGD with the homotopy method (Section 3), come up with a homotopy function to gradually deform a source task distribution into a target one, for both regression and classification tasks (Section 4), and show experimental results for the performance of the method, again for both regression and classification tasks (Section 5). Our homotopy-based method generalizes the widely-used heuristic of fine-tuning networks that were pretrained on a different dataset and is therefore very relevant to the field of deep learning.
>
> Unfortunately, we apparently failed to convey this simple overarching idea in our submission to you. In the new version we just uploaded, we have therefore reworked the abstract, split the introduction into two sections (intro + related work) to allow each of these to be more focused, broke some lengthy sentences into two, and added a list of contributions at the end of the introduction in order to point them out more clearly. We sincerely hope that this new structure of the first part of the paper allows a better overview of our contributions.
>
> Regarding the accessibility of the content, we find the theoretical derivations to be well introduced and structured, and the other reviewers appear to agree (Reviewer #2 mentioned the concepts to be “simple and elegant, and well motivated, and also well introduced”, and Reviewer #3 stated “Overall, the paper is well written, well motivated and well structured. The technical content is also very clear and excellent.”). Unfortunately, given space limit, we had to relegate parts of the derivations to the appendix. As pointed out by Reviewer #3, it would have been nice to move the non-convex local analysis into the main text. If you have any suggestions regarding this issue we would be delighted to take them into consideration. In case you have any doubts on the theoretical derivations and/or the experimental evaluations, we are very happy to discuss these. We would also be very happy to hear more details about your comments in order to help us improve the paper to address them better.
>
> We hope that the rewritten first part of the paper is clearer now, and that based on this, you will reconsider your assessment of the paper. We thank you for your time and effort!

---

### Official Review · AnonReviewer3 · 2019-10-23
**Official Blind Review #3**

**Rating:** 8

**Review:**

Contribution
This paper proposes algorithm for transferring knowledge from easy -to-solved to complex tasks or from already solve to new tasks. It relies on homotopy functions and sequentially solves a sequence of optimization problems where the task distribution is gradually deformed from a source task to the target task. Theoretical guarantees are provided and proven in a strongly convex setting. The main results from the theory show that the distance between the final solution and its optimal are less or equal to  relative to the distance of the initial source solution to its optimum. So a near optimal solution for the source task will lead to near optimal solution for the target task. Regression and Classification experimentations show competitive results compared to  random and warm-start initialization schemes.

Clarity
Overall, the paper is well written, well motivated and well structured. The technical content is also very clear and excellent.
 Minor point: Seems that there is a notation error in proposition G.1 and its proof (i instead of i+1).


Novelty
The novelty in this work seems to be the application of homotopy methods to the transfer learning settings. The mathematical guarantees are also new and may even offer new ways to interpret fine tuning methods that have been so successful in recent literature.

However, given the  non-convexity of DNNs, it seems like the analysis in the non-convex settings and its implications  should be part of the main text.

Experiments:
Overall, the experiments are very insightful but limited since you only show the training loss and the validation performance is not evaluated at all. Other things that would could be  beneficial in better assessing the quality of your method are: comparison to Curriculum learning methods, more in depth analysis of the impact of k, and gamma in both regression and classification settings, and solving  toy convex optimization problems to bridge the gap between theory and application.


Preliminary rating:
* Accept *

**Experience Assessment:**

I have read many papers in this area.

**Review Assessment: Checking Correctness Of Derivations And Theory:**

I assessed the sensibility of the derivations and theory.

**Review Assessment: Checking Correctness Of Experiments:**

I assessed the sensibility of the experiments.

**Review Assessment: Thoroughness In Paper Reading:**

I read the paper at least twice and used my best judgement in assessing the paper.

---

> ### Author Response · Authors · 2019-11-14
> **Thanks for your insightful comments**
>
> Thank you very much for your comments and for taking the time to carefully read the paper. We are really glad that you like our work.
>
> We have refined and polished the non-convex local theoretical analysis that previously was only in the appendix, and moved it to the main text. As you pointed out, this local analysis, since it relies on more realistic assumptions, is closer to the experimental evaluations that we conducted. Moreover, it removes the need for a toy convex evaluation that should have bridged the gap that was previously present between theoretical analysis and experimental evaluations.
>
> Regarding your comment on the notation:
> '' Minor point: Seems that there is a notation error in proposition G.1 and its proof ($i$ instead of $i+1$). ''
> if we understood correctly what the reviewer refers to, we confirm that the indices are used correctly and the mismatch is due to the shift in the homotopy problems, i.e. the change of parameter $\lambda_i$ to $\lambda_{i+1}$.
>
> Regarding the experimental evaluations, we neglected a discussion on the test performances since no theoretical guarantees on the generalization properties are provided. For completeness of the evaluations, as you suggested, we now included this information in the appendix. We did not observe any special trend, but our method seems competitive with the considered baselines also in terms of test performance. We want to underline once again though that we can not formally make any conclusion regarding generalization, since the theoretical analysis does not address this matter.
>
> Regarding a more extensive experimental evaluation of the method, that was also suggested by Reviewer #2, we agree that this might give more insight on the method. However, we believe that that this goes beyond the scope of the paper, whose main focus is to propose a new method, study it theoretically and conduct preliminary numerical evaluations to show the potential of the proposed approach and confirm the theoretical results.

---

### Official Review · AnonReviewer2 · 2019-10-27
**Official Blind Review #2**

**Rating:** 6

**Review:**

Based on homotopy,, the paper describes a more rigorous approach to transfer learning than the so called ‘fine-tuning’ heuristic. Progress in the direction of more principled approaches for transfer learning would be tremendously impactful, since one of the core promises of deep learning is the learning of features, which can be used in different downstream tasks.
Essentially, (if this reviewer understood this correctly) the idea behind this paper works by interpolation between the original task of interest and a potentially easier to optimize surrogate task. Overall, this reviewer found the concept simple and elegant, and well motivated, and also well introduced. However, since this reviewer does not have a formal background in mathematics, they cannot assess the soundness of the proofs.


The paper tests the hypothesis by a simple function approximation regression task, and a classification task to learn to transfer from MNIST to fashion MNIST and MNIST to CIFAR, with promising results. One might argue that a more thorough evaluation would have been desirable, since the claims made by the paper are quite general, and it would have been in the authors’ best interest to present more thorough evidence that their concept works on wider scale of problems, ideally on an NLP task, given the current hype on pre-training with Transformer-based models.




Previous work & citations:

I would recommend to cite Schmidhuber 1987 (Evolutionary principles in self-referential learning) and Hochreiter et al 2001 (Learning to Learn with gradient descent) in the context of Meta learning.
It would be nice to cite Klambauer et al (Self normalizing Networks) in the context of speeding up deep neural network training.
The citations of the VGG paper is currently referenced by first names of the authors, not their last names, I am not sure if this was intended.


**Experience Assessment:**

I do not know much about this area.

**Review Assessment: Checking Correctness Of Derivations And Theory:**

I did not assess the derivations or theory.

**Review Assessment: Checking Correctness Of Experiments:**

I carefully checked the experiments.

**Review Assessment: Thoroughness In Paper Reading:**

I read the paper at least twice and used my best judgement in assessing the paper.

---

> ### Author Response · Authors · 2019-11-14
> **Thank you for your insightful comments**
>
> Thank you very much for taking the time to review the paper and for your comments. We are glad to read that you find the method and our theoretical contribution promising.
>
> Please see  our reply to Reviewer #3 regarding a more extensive experimental evaluation of the method.
>
> Thank you for pointing out the typo in the citation of the VGG paper.
> We have also included the citations that you suggested for meta-learning and in the context of speeding up the training of deep networks.

---

> > ### Comment · AnonReviewer2 · 2019-11-14
> > **thank you for your reply**
> >
> > This reviewer agrees that further experimentation can also be conducted in follow up work.

---

### Decision · Program_Chairs · 2019-12-19

**Decision:**

Accept (Poster)

**Comment:**

This paper presents a theoretically motivated method based on homotopy continuation for transfer learning and demonstrates encouraging results on FashionMNIST and CIFAR-10. The authors draw a connection between this approach and the widely used fine-tuning heuristic. Reviewers find principled approaches to transfer learning in deep neural networks an important direction, and find the contributions of this paper an encouraging step in that direction. Alongside with the reviewers, I think homotopy continuation is a great numerical tool with a lot of untapped potentials for ML applications, and I am happy to see an instantiation of this approach for transfer learning. Reviewers had some concerns about experimental evaluations (reporting test performance in addition to training), and the writing of the draft. The authors addressed these in the revised version by including test performance in the appendix and rewriting the first parts of the paper. Two out of three reviewers recommend accept. I also find the homotopy analysis interesting and alongside with majority of reviewers, recommend accept. However, please try to iterate at least once more over the writing; simply long sentences and make sure the writing and flow are, for the camera ready version.